# Mechanisms of Ohmic Contact Formation of Ti/Al-Based Metal Stacks on p-Doped 4H-SiC

**DOI:** 10.3390/ma15010050

**Published:** 2021-12-22

**Authors:** Matthias Kocher, Mathias Rommel, Paweł Piotr Michałowski, Tobias Erlbacher

**Affiliations:** 1Fraunhofer Institute for Integrated Systems and Device Technology (IISB), Schottkystrasse 10, 91058 Erlangen, Germany; mathias.rommel@iisb.fraunhofer.de (M.R.); tobias.erlbacher@iisb.fraunhofer.de (T.E.); 2Łukasiewicz Research Network—Institute of Microelectronics and Photonics, Aleja Lotników 32/46, 02-668 Warsaw, Poland; pawel.michalowski@imif.lukasiewicz.gov.pl; 3Chair of Electron Devices, Friedrich-Alexander-University Erlangen-Nuremberg, Cauerstraße 6, 91058 Erlangen, Germany

**Keywords:** 4H-SiC, ohmic contact, SIMS, Ti_3_SiC_2_, simulation

## Abstract

Ohmic contacts on p-doped 4H-SiC are essential for the fabrication of a wide range of power electron devices. Despite the fact that Ti/Al based ohmic contacts are routinely used for ohmic contacts on p-doped 4H-SiC, the underlying contact formation mechanisms are still not fully understood. TLM structures were fabricated, measured and analyzed to get a better understanding of the formation mechanism. SIMS analyses at the Ti_3_SiC_2_-SiC interface have shown a significant increase of the surface near Al concentration. By using numerical simulation it is shown that this additional surface near Al concentration is essential for the ohmic contact formation.

## 1. Introduction

Silicon carbide (SiC) is predestined for the fabrication of high power, high temperature and high frequency semiconductor devices, due to its outstanding properties. Despite the commercial availability of SiC power devices, like vertical MOSFETs or Schottky barrier diodes, some topics are not fully understood yet and need further investigations. One major topic is the understanding of ohmic contact formation mechanism on p-doped 4H-SiC [1], due to its importance in the fabrication of bipolar power devices, like pin-diodes or Insulated Gate Bipolar Transistors (IGBTs).

In order to fabricate reliable and low ohmic contacts on p-doped 4H-SiC, various metals and metal stacks have been investigated [1,2]. Due to rather low obtained contact resistivities Ti/Al based metal stacks have become a quasi-standard for ohmic contacts on p-type SiC [1,2,3]. Numerous studies with different Ti/Al ratios and stacking sequences as well as varying Al surface concentrations fabricated by epitaxial growth or implantation were done in order to fabricate low-ohmic contacts on p-doped 4H SiC [3,4,5,6,7,8,9,10,11]. These studies have shown the possibility of fabricating ohmic contacts with a specific contact resistance $\rho_{C}$ down to $10^{-6}\;{\Omega \mathrm{cm}}^{2}$ [2,6].

It was also shown that the formation of Ti/Al based low ohmic contacts correlates with the appearance of Ti_3_SiC_2_ on the 4H-SiC surface. Several investigations in literature have revealed an epitaxial growth of Ti_3_SiC_2_ during contact formation [3,8,11,12,13,14,15,16]. Maeda et al. have described that the epitaxial growth of Ti_3_SiC_2_ consists of two separate reactions (see Equations (1) and (2)) [14].
(1)$$Ti+3Al=TiAl_{3}\;\mathrm{at}\;959\;K$$
(2)$$2SiC+3TiAl_{3}=Ti_{3}SiC_{2}+9Al+Si\;\;\mathrm{at}\;1270\;K$$

Ti_3_SiC_2_ formation is key for ohmic contact formation mechanism [1,2,3,6,8,12,17]. Notwithstanding these results, the formation mechanism of Ti/Al based ohmic contacts on p-doped 4H-SiC is not fully understood [1]. Therefore, this work investigates the underlying formation mechanism and sets up a theory for the contact formation mechanism by analyzing and simulating the Ti/Al based ohmic contacts interface.

## 2. Materials and Methods

Transfer length method (TLM) structures with increasing pad distances (30 µm to 480 µm) were fabricated on n^−^ doped 4H-SiC epitaxial layers in order to investigate the contact formation mechanism of Ti/Al based ohmic contact. Al implantation and subsequent high temperature annealing was used to create samples with p^+^ doped regions and Al surface concentrations between 3.3 × 10^18^ cm^−3^ and 5.0 × 10^19^ cm^−3^. After depositing and structuring a SiO_2_ passivation layer, a metal stack consisting of 80 nm Ti and 300 nm Al was deposited and patterned using a lift-off process. Subsequently ohmic contact formation was done by rapid thermal annealing (RTA) at 980 °C in Ar atmosphere. Finally, metal pads for electrical measurements were deposited and structured. A more detailed description of the fabrication process of sample C can be seen elsewhere [18]. Table 1 gives an overview of the fabricated sets of samples and their fabrication parameters.

By using 4-point I-V-measurements (Keithley 4200 Parameter Analyzer, Keithley Instruments, USA) at different temperatures (300 K to 450 K), the fabricated TLM structures were electrically characterized and the sheet resistance $R_{sh}$, the contact resistance $R_{C}$ as well as the specific contact resistance $\rho_{C}$ were determined in the given temperature range.

Focused ion beam (FIB) (Helios Nanolab 600, FEI, USA) as well as transmission electron microscopy (TEM) analysis were done to determine the thickness of the Ti_3_SiC_2_ layer (approx. 100 nm) as well as its stoichiometry.

## 3. Results

### 3.1. Sheet Resistance and Determination of the Acceptor Ionization Energy

All fabricated samples show ohmic behavior across all measurement temperatures. The sheet resistance $R_{sh}$ was used to determine the acceptor ionization energies $\Delta E_{A}$ of the fabricated samples. Equation (3) (together with Equations (4)–(8)) can be used to describe the sheet resistance $R_{sh}$ of a semiconductor, where q indicates the elementary charge, t the thickness of the semiconductor layer, p and n the hole and electron concentrations, $\mu_{p}$ and $\mu_{n}$ the hole and electron mobilities, respectively.
(3)$$R_{sh}={\left(q\int_{0}^{t}\left(\mu_{n}\left(x\right)n\left(x\right)+\mu_{p}\left(x\right)p\left(x\right)\right)dx\right)}^{-1}$$

Equations (4)–(6) were used to calculate the hole and electron mobility and their respective temperature dependence, where $\mu_{const}$ describes the mobility due to phonon scattering, $\mu_{dop}$ the doping dependent mobility degradation, T the temperature, $N_{D}$ the donor concentration and $N_{A}$ the acceptor concentration (all other parameters and their values are shown in Table A1 [19].
(4)$$\mu ={\left(\mu_{const}^{-1}+\mu_{dop}^{-1}\right)}^{-1}$$
(5)$$\mu_{const}=\mu_{L}{\left(\frac{T}{300K}\right)}^{-\xi }$$
(6)$$\mu_{dop}=A_{min}{\left(\frac{T}{300K}\right)}^{\alpha_{m}}+\frac{A_{d}{\left(\frac{T}{300K}\right)}^{\alpha_{d}}}{1+{\left(\frac{N_{A}+N_{D}}{A_{N}{\left(\frac{T}{300K}\right)}^{\alpha_{N}}}\right)}^{A_{a}{\left(\frac{T}{300K}\right)}^{a_{a}}}}$$

Equations (7) and (8) were used to describe the carrier ionization, where $N_{D}^{+}$ describes the ionized donor concentration and $N_{A}^{-}$ describes the ionized acceptor concentration (all other parameters and their values are shown in Table A2 [1]. Here, a negligible carrier compensation ($p\approx N_{A}^{-}$) was assumed at first.
(7)$$n\approx N_{D}^{+}=\frac{\eta_{n}}{2}\left(\sqrt{1+\frac{4N_{D}}{\eta_{n}}}-1\right)\;\mathrm{with}\;\begin{array}{c} \eta_{n}=\frac{N_{C}}{g_{D}}exp\left(-\frac{\Delta E_{D}}{kT}\right)\\ N_{C}=N_{C,300K}{\left(\frac{T}{300K}\right)}^{1.5} \end{array}$$
(8)$$p\approx N_{A}^{-}=\frac{\eta_{p}}{2}\left(\sqrt{1+\frac{4N_{A}}{\eta_{p}}}-1\right)\;\mathrm{with}\;\begin{array}{c} \eta_{p}=\frac{N_{V}}{g_{A}}exp\left(-\frac{\Delta E_{A}}{kT}\right)\\ N_{V}=N_{V,300K}{\left(\frac{T}{300K}\right)}^{1.5} \end{array}$$

In Figure 1a the normalized average measured sheet resistances and the associated standard error of all sets of samples with an implanted Al surface concentration of 5 × 10^19^ cm^−3^ are shown. It can be seen that all sets of samples show similar temperature dependent behavior despite differences in sheet resistance values (see inset in Figure 1a).

The associated acceptor ionization energies $\Delta E_{A}$ were determined by fitting the theoretical sheet resistance to the measured ones. The theoretical sheet resistance was determined by using the simulated implantation profile and assuming 100% activation of the dopants.

The determined effective acceptor ionization energies $\Delta E_{A}$ as well as the theoretical acceptor ionization energies (see Equation (9) with $\Delta E_{A,0}$ = 0.265 eV [19,20]) are shown in Figure 1b. It can be seen that the determined acceptor ionization energies differ significantly from the theoretical ones, which can be explained by a significant amount of carrier compensation centers (see Equation (8)). As discussed in Section 3.3, these compensation centers might be modelled by donor-like defects that trap free holes.
(9)$$\Delta E_{A}=\Delta E_{A,0}-\sqrt[{3}]{N_{D}+N_{A}}$$

### 3.2. Determination of Schottky Barrier Height

Based on the Thermionic Field Emission (TFE) model [21], Equation (10) can be used to determine the Schottky barrier height $\phi_{B}$, where k describes the Boltzmann constant, h the Planck constant, $m^{*}$ the effective tunneling mass (here 0.91 electron masses [22,23,24]), $\epsilon_{0}$ the vacuum permittivity and $\epsilon_{S}$ the relative permittivity of 4H-SiC (here 9.7 [4,22,23]).
(10)$$\rho_{C,TFE}\propto \frac{q\phi_{B}}{E_{00}\coth\left(\frac{E_{00}}{kT}\right)}\;with\;E_{00}=\frac{qh}{2\pi }\sqrt{\frac{p}{m^{*}\epsilon_{0}\epsilon_{S}}}$$

The Schottky barrier height $\phi_{B}$ itself can be calculated by Equation (11), where $E_{g}$ describes the bandgap of the semiconductor, $\phi_{M}$ the metal workfunction of the ohmic contact material, $\chi_{S}$ the electron affinity of the semiconductor and $V_{i}$ the built-in voltage [21,25]. It can be seen that the Schottky barrier height decreases slightly with increasing $N_{A}^{-}$.
(11)$$\phi_{B}=\frac{E_{g}}{q}-\left(\phi_{M}-\chi_{S}\right)-\sqrt[{4}]{\frac{q^{3}N_{A}^{-}V_{i}}{8\pi^{2}\epsilon_{0}^{3}\epsilon_{S}^{3}}}$$

Figure 2a shows the normalized average specific contact resistances of all sets of samples with an implanted Al surface concentration of 5 × 10^19^ cm^−3^. It can be observed that all sets of samples show quite similar temperature dependent behavior despite different absolute values of the specific contact resistances (see inset of Figure 2a) and despite the deviation of sample B at temperatures higher than 375 K (indicated by the open red squares). The origin of these deviations is not fully understood. Therefore, these values are not used further.

By fitting the theoretical specific contact resistance to the measured ones, the Schottky barrier heights $\phi_{B}$ were determined (see Equation (10)). Figure 2b shows the determined Schottky barrier heights from this work and compares them with Schottky barrier heights known from literature. It can be seen that the determined Schottky barrier heights increase with increasing Al surface concentration, which is in contrast to the theoretically predicted decreasing of the Schottky barrier height with increasing Al surface concentration (see Equation (11)).

In order to investigate this contradiction Secondary Ion Mass Spectrometry (SIMS) analyses were carefully done at the 4H-SiC/Ti_3_SiC_2_-interface of sample A by using a CAMECA IMS SC Ultra SIMS tool which allows a sub-nm resolution [28,29]. The sub-nm depth resolution was achieved for O^2+^ primary ions with an impact energy of 250 eV. The Al concentration was calibrated using a reference sample consisting of a SiC substrate implanted with Al ions with an energy of 100 keV and a dose of 10^14^ cm^−2^.

Figure 3a shows the measured Al concentration, the measured Ti and Si counts per second (CPS) as well as the implanted Al profile. While no Al could be detected in the Ti_3_SiC_2_ layer, the Al concentration at the 4H-SiC-Ti_3_SiC_2_-interface is significantly increased. This additional Al concentration decreases within approx. 3 nm from a peak concentration of approx. 10^21^ cm^−3^ to the implanted Al concentration (5 × 10^19^ cm^−3^). Furthermore no significant amount of Ti could be detected in the 4H-SiC layer.

This increase of the Al concentration can be explained by a diffusion of Al during Ti_3_SiC_2_ formation. The total resulting Al profile can be approximated by the superposition of the implanted Al profile $N_{Al,impl.}$ and the increase of the Al concentration at the 4H-SiC/Ti_3_SiC_2_-interface. Equation (12) describes this superposition by using the implanted Al profile $N_{Al,impl.}$, the diffused Al dose during high temperature annealing $N_{Al,dose}$ and the associated diffusion length $L_{Al,diff}$ [25]. Figure 3b shows the approximation as well as the associated parameters. It can be seen, that the approximation fits very well with the measured data.
(12)$$N_{Al}\left(x\right)=N_{Al,impl.}\left(x\right)+\frac{2\;N_{Al,dose}}{L_{Al,diff}\sqrt{\pi }\;}\exp\left(-\frac{x^{2}}{L_{Al,diff}^{2}}\right)$$

### 3.3. Numerical Simulation

To investigate the influence of the surface-near increased Al concentration as well as the temperature dependent behavior of sample A a numerical simulation model was developed by using Sentaurus TCAD (Version O_2018.06). Figure 4 shows the scheme of the used simulation model. This model includes Ti_3_SiC_2_ based ohmic contact pads with a height of 100 nm and a pad distance d between the ohmic contacts. The model includes further a homogeneous n^−^-doped 4H-SiC epitaxial layer, a p^+^-doped region and a p^++^-doped region beneath the ohmic contacts. The p^+^-doped region was created by using a Monte Carlo simulation of the implanted Al profiles. The surface near p^++^-doped region was created by adding a Gaussian distributed Al profile with diffused Al dose $N_{Al,dose}$ and the associated diffusion length $L_{Al,diff}$. Furthermore the model assumes complete activation of the Al atoms and takes account of incomplete ionization as well as doping and temperature dependent carrier mobility.

An additional virtual N profile was added in order to model the concentration of carrier compensation centers. The distribution of this additional virtual N profile is identical with the distribution of the implanted Al profile and can be scaled by using the compensation ratio $f_{comp}$. Due to these additional donor atoms the concentration of free holes can be reduced similarly to compensation by donor-like traps which increases the associated sheet resistance and allows to fit $R_{sh}$.

Using this numerical simulation model, I-V characteristics depending on the diffused Al dose $N_{Al,dose}$, the associated diffusion length $L_{Al,diff}$ and the compensation ratio $f_{comp}$ for each temperature and each pad distance d can be obtained. This allows to simulate I-V characteristics for TLM structures at different temperatures.

Based on these I–V characteristics the sheet resistance $R_{sh}$ and the contact resistance $R_{C}$ of the modelled TLM structures were determined. Figure 5a compares the sheet resistance $R_{sh}$, Figure 5b compares the contact resistance $R_{C}$ determined from the electrical measurement data with simulated ones of sample A. It can be seen that both fits are in decent agreement with the measurements. It should be mentioned here that no adjustments on the parameters $N_{Al,dose}$ and $L_{Al,diff}$ determined by the SIMS analysis were necessary.

The determined compensation ratio $f_{comp}$ is dependent on the temperature as shown in Figure 6. It can be seen that $f_{comp}$ increases slightly with increasing temperature from 8.3% at 300 K to 10.5% at 450 K which fits to temperature independent compensation ratios known from literature (10% to 27%) [30,31,32]. This temperature dependence might be explained by the fact that the ionization energy from nitrogen differs from the ionization energy of the actual compensation centers.

## 4. Discussion

Due to the well-fitted simulation results, it can be concluded that the numerical simulation model is suitable to describe Ti_3_SiC_2_ based ohmic contacts on p-doped 4H-SiC temperature dependent. Considering the fact that the simulation model does not show ohmic behavior when not using the surface near Al profile it can be further concluded that the surface near Al profile is essential for the ohmic contact formation. Based on these results it is possible to propose a theory regarding the formation mechanism of Ti/Al based ohmic contact on p-doped 4H-SiC and the role of Ti_3_SiC_2_ during contact formation.

During the Ti_3_SiC_2_ formation a certain amount of Al diffuses in the SiC surface via lattice places and increases the surface near Al concentration significantly. This increase of the surface near Al concentration can significantly decrease the specific resistance $\rho_{C}$ (see Equation (10)) and is therefore the key in the ohmic contact formation.

Further investigations are necessary to verify this model and to obtain a better understanding of the conditions leading to the ohmic contact formation under various process conditions. Nevertheless, the fundamental effects are becoming accessible for process integration and process modelling.

## Figures and Tables

**Figure 1 materials-15-00050-f001:**
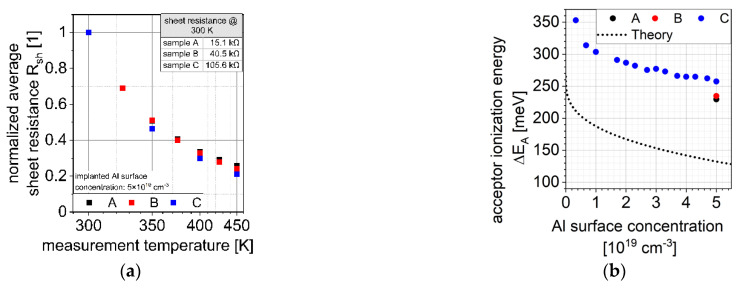
(**a**) Measurement temperature dependence of normalized average sheet resistance $R_{sh}$; (**b**) Determined effective acceptor ionization energies $\Delta E_{A}$.

**Figure 2 materials-15-00050-f002:**
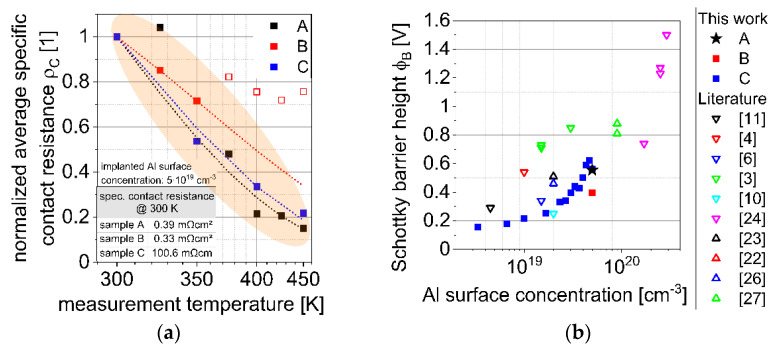
(**a**) Normalized average specific contact resistance (**b**) Determined Schottky barrier heights form this work and literature. Ohmic contacts from literature fabricated on epitaxial regions are indicated with ∇ ([3,4,6,10,11,24]), ohmic contacts from literature fabricated on implanted regions are indicated with Δ ([22,23,26,27]).

**Figure 3 materials-15-00050-f003:**
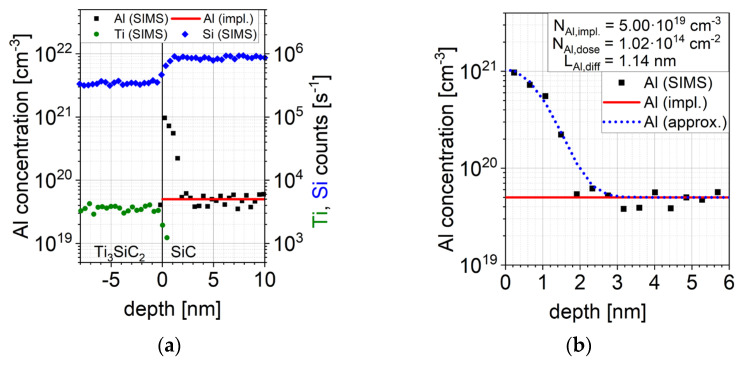
(**a**) Measured SIMS profiles of Al, Ti and Si on the Ti_3_SiC_2_-SiC interface (**b**) Approximation of the measured Al profile.

**Figure 4 materials-15-00050-f004:**
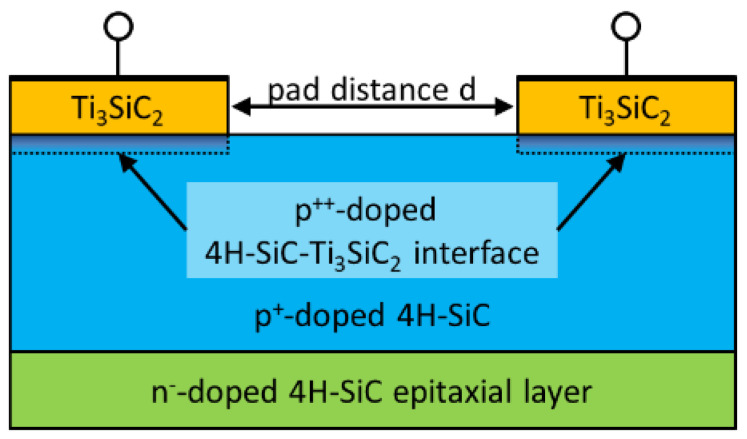
Scheme of the simulation model.

**Figure 5 materials-15-00050-f005:**
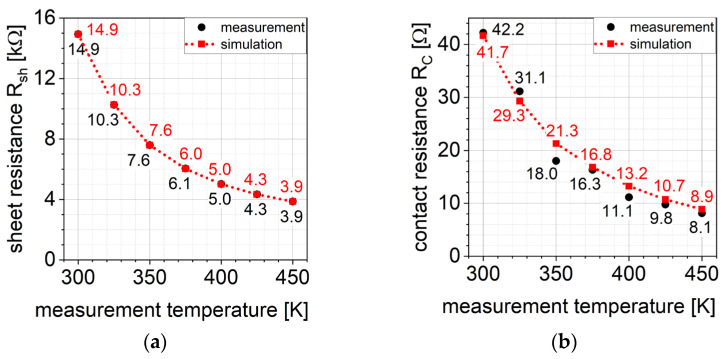
(**a**) Measured and simulated sheet resistance (**b**) Measured and simulated contact resistance.

**Figure 6 materials-15-00050-f006:**
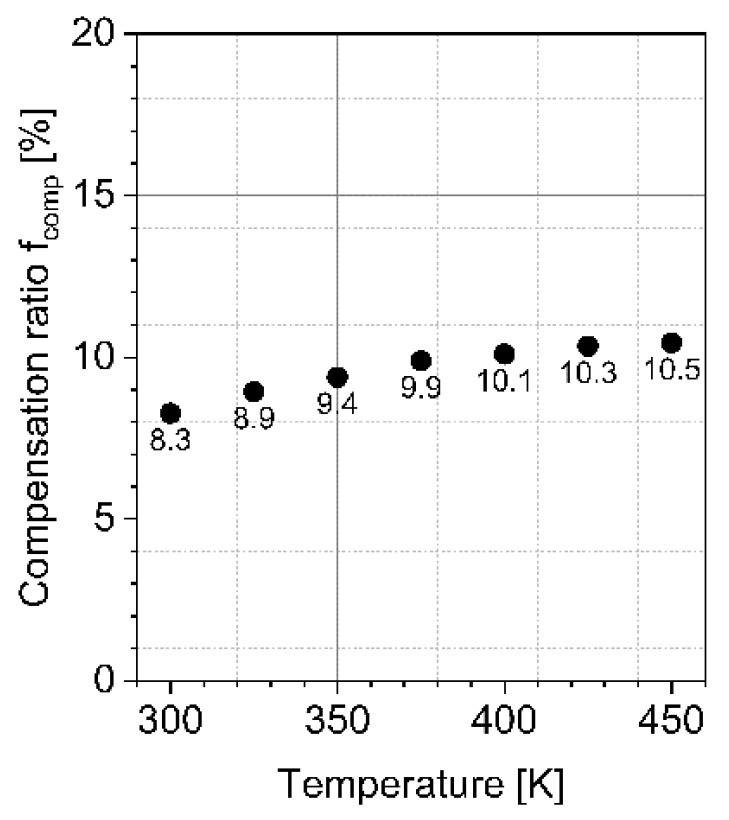
Temperature dependence of determined compensation ratio.

**Table 1 materials-15-00050-t001:** Parameters of the fabricated sets of samples.

	A	B	C
Implanted Al surface concentration [10^19^ cm^−3^]	5.0	5.0	0.33 to 5.0 (14 different Al surface conc.)
Implanted Al dose [10^14^ cm^−2^]	9.0	6.0	0.34 to 5.1 (14 different Al doses)
High temperature implantation annealing	1700 °C, 30 min, Ar atmosphere		

## Data Availability

The datasets generated and analysed during the current study are available from the corresponding author on reasonable request.

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
