# Peer review of "Mechanisms of Ohmic Contact Formation of Ti/Al-Based Metal Stacks on p-Doped 4H-SiC"

_materials, 2021, doi:10.3390/ma15010050_

Round 1

Reviewer 1 Report

This paper reported the ohmic contact of Ti/Al on p-doped 4H-SiC with Al implantation. The ohmic contact formation mechanism after annealing was proposed based on the experiment and numerical simulation. The results seem reasonable. However, there are some points to be improved.

  1. In the experimental, why sample A and B with same Al surface concentration were used? It seems no comparison between them. Additionally, why the concentration of sample C is not a certain value, but 0.33 to 5.0?
  2. Authors have pointed that the Schottky barrier heights increase with increasing Al surface concentration, which is different from the common sense. Could you give the reason? By the way, could you analyze why the Schottky barrier height is lower than the literatures?
  3. Although authors explained the Al concentration by a diffusion of Al during Ti3SiC2 formation, more analysis on the diffusion information such as diffusion coefficients of Al in the Al/Ti and implanted Al should be given.
  4. It is suggested to provide the SIMS results of Al, Ti and Si before the annealing process, which could give the more direct evidence, or the surface phase composition before and after the annealing.
  5. In Fig.2(a), the unit of the specific contact resistance in the vertical coordinate should be added.

Author Response

1.    In the experimental, why sample A and B with same Al surface concentration were used? It seems no comparison between them.
--> Thank you for pointing out this inaccuracy. Implanted Al dose was added to show the difference between samples A and B.
Additionally, why the concentration of sample C is not a certain value, but 0.33 to 5.0?
--> Thank you for pointing out this inaccuracy. A more detailed explanation has been added.

2.    Authors have pointed that the Schottky barrier heights increase with increasing Al surface concentration, which is different from the common sense. Could you give the reason? By the way, could you analyze why the Schottky barrier height is lower than the literatures? 
--> With a given specific contact resistance not considering the increase of the surface near Al concentration will lead to miscalculation of the barrier height. The more the calculated Al surface concentration differs from the real one the lower the Schottky barrier height will become.

3.    Although authors explained the Al concentration by a diffusion of Al during Ti3SiC2 formation, more analysis on the diffusion information such as diffusion coefficients of Al in the Al/Ti and implanted Al should be given.
--> Unfortunately, the authors cannot provide further information due to the fact that analyzing diffusion coefficients in a proper manner will require further analysis on further sets of samples.

4.    It is suggested to provide the SIMS results of Al, Ti and Si before the annealing process, which could give the more direct evidence, or the surface phase composition before and after the annealing.
--> Unfortunately, no SIMS analyses were performed before annealing.

5.    In Fig.2(a), the unit of the specific contact resistance in the vertical coordinate should be added.
--> Unfortunately, the authors do not understand your point. Due to the fact that the vertical axis shows the a normalized value the unit is "1" as labelled.

Reviewer 2 Report

This paper investigates the electrical behavior of Ti/Al-based contacts formed on Al-doped 4H-SiC mainly by forming TLM structures and fitting the measured data with TCAD simulations. Additional information is obtained from SIMS. The paper is well written and logically structured. The results are of interest for the SiC device community, but the paper needs additional information and clarifications along the comments below, before it can be published.

Line 18 in Abstract (and also at other places), please clarify that it is the Al concentration in the 4H-SiC that has increased. Maybe “…surface-near Al concentration in the 4H-SiC …”

In the introduction (line 49) it is stated that the paper “sets up a theory for the contact formation mechanism …”, which is not really the case. Maybe something like “… suggests an enhanced Al concentration in the top SiC layer to be the reason for …”

The sentence on line 31-32, “Due to rather …” needs re-phrasing. Maybe remove “with”.

What is the n-doping level in the epi? If it is several orders of magnitude lower than the p-type implanted layer, the contribution of electron current is probably not needed in the simulations.

Please, provide more information on the implantations: multiple energies, sacrificial layer to put peak Al concentration at the surface, polishing after annealing, how is the thickness estimated (considering the slow decay of concentration after the deepest implant), room temperature, …

The Al implantation was simulated by Monte Carlo method. Please provide a reference to the program.

Motivate your choice of degeneracy factors, g_D=g_A=1 (underscore means subscript). More often you find 2 and 4, respectively.

Incomplete ionization and bandgap narrowing are very sensitive to the ionization energy of the Al acceptor. It is fairly agreed that it is positioned about 0.20+/- 0.01 eV above the valence band, with some difference for hexagonal and cubic sites. You use another value, deltaE_A0=0.265, which is very much higher. I think you need to clarify and motivate this in more detail, for instance a proper reference to Eq. 9 (not the User’s guide).

The SIMS results are crucial for the interpretation of the results. However, SIMS is highly unpredictable at interfaces due to varying ionization rates for an element in different matrixes. Please, make some statement about why you trust your SIMS Al profile. (I don’t think that you manage to fit a diffusion profile is sufficient.) Also include a reference to Eq. 12 and the value for the diffusion length for Al in the (growing!) Ti-Si-C layer.

Regarding the SIMS data in Fig. 3a, it is a bit confusing with the y-scales Al conc and Ti, Si counts, e.g. in the left half of the figure you have Al-conc. on the y-axis, but you plot only Si and Ti.

Did you only convert the Al-profile from counts to cm^-3 (^ is superscript)? Please, explain better or consider a new figure. Also, it would be excellent if the Si, C and Ti counts could be converted to make sure that you have a stoichiometric MAX phase.

Why was 100 nm chosen for the thickness of the Ti-Si-C layer? Is the Ti fully “eaten up” by the SiC? Does the Al on top of the contact in Fig. 4 not add any resistance, or barrier?

I am hesitant to using N_D+ as a “substitute” for traps (N_T).

Author Response

Line 18 in Abstract (and also at other places), please clarify that it is the Al concentration in the 4H-SiC that has increased. Maybe “…surface-near Al concentration in the 4H-SiC …”
--> Thank you for pointing this out. Clarifications have been added.

In the introduction (line 49) it is stated that the paper “sets up a theory for the contact formation mechanism …”, which is not really the case. Maybe something like “… suggests an enhanced Al concentration in the top SiC layer to be the reason for …”
--> Thank you for pointing out your concerns. However, the authors believe that the conclusions presented here, are more than just suggestions.

The sentence on line 31-32, “Due to rather …” needs re-phrasing. Maybe remove “with”.
--> Thank you for pointing out this improvement. Changes have been done accordingly.

What is the n-doping level in the epi? If it is several orders of magnitude lower than the p-type implanted layer, the contribution of electron current is probably not needed in the simulations.
--> Unfortunately, the authors do not fully understand your point. The n-doping level in the epitaxial layer is between 5e15 1/cm³ and 1.6e16 1/cm³ depending on the sample.

Please, provide more information on the implantations: multiple energies, sacrificial layer to put peak Al concentration at the surface, polishing after annealing, how is the thickness estimated (considering the slow decay of concentration after the deepest implant), room temperature, …
--> Thank you for pointing this out. The authors believe that the manufacturing process is not essential to understand the publication. A detailed description of the fabrication process of sample C can be seen here.: DOI: 10.4028/www.scientific.net/MSF.924.393

The Al implantation was simulated by Monte Carlo method. Please provide a reference to the program.
--> Both types of simulations (process and device) were done by using Sentaurus TCAD. 

Motivate your choice of degeneracy factors, g_D=g_A=1 (underscore means subscript). More often you find 2 and 4, respectively.
--> Unfortunately, the authors do not understand your point. As shown in Table 3 both values were chosen to be 2 and 4, respectively. Maybe you mix it up with the unit which is 1.

Incomplete ionization and bandgap narrowing are very sensitive to the ionization energy of the Al acceptor. It is fairly agreed that it is positioned about 0.20+/- 0.01 eV above the valence band, with some difference for hexagonal and cubic sites. You use another value, deltaE_A0=0.265, which is very much higher. I think you need to clarify and motivate this in more detail, for instance a proper reference to Eq. 9 (not the User’s guide).
--> Thank you for pointing out this improvement. Additional reference for deltaE_A0=0.265 has been added.

The SIMS results are crucial for the interpretation of the results. However, SIMS is highly unpredictable at interfaces due to varying ionization rates for an element in different matrixes. Please, make some statement about why you trust your SIMS Al profile. (I don’t think that you manage to fit a diffusion profile is sufficient.) Also include a reference to Eq. 12 and the value for the diffusion length for Al in the (growing!) Ti-Si-C layer.
--> Unfortunately, the authors do not fully understand your doubts. The references from Paweł Michałowski prove the possibility of sub-nm SIMS analysis. Fig. 3(b) shows the diffusion profile as well as the diffusion length. A reference to Eq. 12 has been added.

Regarding the SIMS data in Fig. 3a, it is a bit confusing with the y-scales Al conc and Ti, Si counts, e.g. in the left half of the figure you have Al-conc. on the y-axis, but you plot only Si and Ti.
--> Unfortunately, the authors do not fully understand your point. 

Did you only convert the Al-profile from counts to cm^-3 (^ is superscript)? Please, explain better or consider a new figure. Also, it would be excellent if the Si, C and Ti counts could be converted to make sure that you have a stoichiometric MAX phase. 
--> The authors do not fully understand your point. As usual using Al calibration allows to convert Al counts per second into a concentration. The stoichiometry was proven by using TEM analysis.

Why was 100 nm chosen for the thickness of the Ti-Si-C layer? Is the Ti fully “eaten up” by the SiC? Does the Al on top of the contact in Fig. 4 not add any resistance, or barrier? 
--> Thank you for pointing out this improvement. FIB measurement results have been added. Using a 4 point measurement allows to eliminate the Al resistance during measurement.

I am hesitant to using N_D+ as a “substitute” for traps (N_T). 
--> Thank you for pointing out this improvement. The authors decided not to change the symbol.

Reviewer 3 Report

The Authors developed numerical model of Ti3SiC2/4H-SiC contacts formation and simulate temperature dependence of the contact resistance of Ti3SiC2/4H-SiC. The model explains formation of ohmic contacts by the diffusion of the Al into SiC surface. The results of simulation supported by sub-nm SIMS analysis of sub-surface Al profile and experimental measurements of termal denended contact resistance by TLM aproach.

For further improvement of the manuscript the authors should consider following comments:

  1. Line 29. Please, specify abreviation IGBTs
  2. Line 53. The configuration of experimental structure could be explained in more detailes. Was it TLM configuration of contacts with variable distance between contact pads or 4-point configuration with fixed distance between contact pads.
  3. Line 88. Not all of variables used in equations are defined in the text.
  4. Line 109. Some sugestions or speculations about the origin on the compensation centers could be provided.
  5. Line 135. Was the Schottky barier height (SBH) calculated in frame of TFE theory?
  6. Line 139. Could authors provide any sugestions about origin of contradiction between the experimental and teoretical dependences SBH vs. Al concentration.
  7. Line 214. Could authors make any assumpion why Al decreases the contact resistance?

Author Response

1.    Line 29. Please, specify abreviation IGBTs
--> Done. Thank you for pointing out this inaccuracy.

2.    Line 53. The configuration of experimental structure could be explained in more detailes. Was it TLM configuration of contacts with variable distance between contact pads or 4-point configuration with fixed distance between contact pads.
--> Thank you for pointing this out. The pad distances have been added to the text.

3.    Line 88. Not all of variables used in equations are defined in the text.
--> Thank you for pointing this out. You are totally right that not all variables are defined in text, the “missing” variables as well as their value and unit are shown in Table 2 and Table 3.

4.    Line 109. Some sugestions or speculations about the origin on the compensation centers could be provided.
--> Due to the fact that the origin of compensation centers is not fully understood the authors prefer to not adding speculations here, because this will just expand the publication.

5.    Line 135. Was the Schottky barier height (SBH) calculated in frame of TFE theory? 
--> Yes, the SBH was calculated by using Eq. 10. Also added to the text.

6.    Line 139. Could authors provide any sugestions about origin of contradiction between the experimental and teoretical dependences SBH vs. Al concentration. 
--> With a given specific contact resistance not considering the increase of the surface near Al concentration will lead to miscalculation of the barrier height. The more the calculated Al surface concentration differs from the real one the lower the Schottky barrier height will become.

7.    Line 214. Could authors make any assumpion why Al decreases the contact resistance? 
--> Eq. 10 can be used to determine the contact resistance in dependence of the hole concentration. By increasing the Al concentration and assuming complete activation of the additional Al atoms the hole concentration will increase and therefore the contact resistance will decrease.

Reviewer 4 Report

The manuscript is devoted to the process of forming ohmic contacts to a 4H-SiC device. Though quite a lot of papers published on the same topic, the work is well organized and presents both experimental an simulation results complementing each other. I have few questions to be clarified:
1.    Apart from simulation model, it would be better to see the initial experimental structures or corresponding drawings in the Materials and Methods section with the sample structure dimensions.
2.    The surface concentration is specified in Materials and Methods in cm-3. In this case, either the layer thickness is required or the 2D surface concentration must be shown.
3.    How did the authors get this concentration? Was it calculated? Is there any uncertainty? This becomes clearer only in Figure 3, but I would advise to put a short description of SIMS to the section 2.
4.    It would be nice to show references in the legend of the Fig. 2b instead of letters a, b, c, …
I would suggest the manuscript to be published after minor improvements.

Author Response

1.    Apart from simulation model, it would be better to see the initial experimental structures or corresponding drawings in the Materials and Methods section with the sample structure dimensions.
--> Thank you for pointing out this improvement. The sample structure dimensions were added to the text.

2.    The surface concentration is specified in Materials and Methods in cm-3. In this case, either the layer thickness is required or the 2D surface concentration must be shown.
--> The authors do not fully understand your point. The term “Al surface concentration” decribes the concentration at da depth of 0 nm.

3.    How did the authors get this concentration? Was it calculated? Is there any uncertainty? This becomes clearer only in Figure 3, but I would advise to put a short description of SIMS to the section 2.
--> The authors do not fully understand your point.

4.    It would be nice to show references in the legend of the Fig. 2b instead of letters a, b, c, …
--> Thank you for pointing this out. The authors decide not change Fig. 2(b).